# Non-conditioned bone marrow chimeric mouse generation using culture-based enrichment of hematopoietic stem and progenitor cells

Kiyosumi Ochi[1,5], Maiko Morita [1,5], Adam C. Wilkinson[2], Atsushi Iwama[3] & Satoshi Yamazaki[1,4✉]

Bone marrow (BM) chimeric mice are a valuable tool in the field of immunology, with the genetic manipulation of donor cells widely used to study gene function under physiological and pathological settings. To date, however, BM chimera protocols require myeloablative conditioning of recipient mice, which dramatically alters steady-state hematopoiesis. Additionally, most protocols use fluorescence-activated cell sorting (FACS) of hematopoietic stem/progenitor cells (HSPCs) for ex vivo genetic manipulation. Here, we describe our development of cell culture techniques for the enrichment of functional HSPCs from mouse BM without the use of FACS purification. Furthermore, the large number of HSPCs derived from these cultures generate BM chimeric mice without irradiation. These HSPC cultures can also be genetically manipulated by viral transduction, to allow for doxycycline-inducible transgene expression in donor-derived immune cells within non-conditioned immuno-competent recipients. This technique is therefore expected to overcome current limitations in mouse transplantation models.

[1] Division of Stem Cell Biology, Center for Stem Cell Biology and Regenerative Medicine, The Institute of Medical Science, The University of Tokyo, Tokyo, Japan. [2] MRC Molecular Hematology Unit, MRC Weatherall Institute of Molecular Medicine, University of Oxford, Radcliffe Department of Medicine, University of Oxford, Oxford, UK. [3] Division of Stem Cell and Molecular Medicine, Center for Stem Cell Biology and Regenerative Medicine, The Institute of Medical Science, The University of Tokyo, Tokyo, Japan. [4] Laboratory of Stem Cell Therapy, Faculty of Medicine, University of Tsukuba, Ibaraki, Japan. [5]These authors contributed equally: Kiyosumi Ochi, Maiko Morita. ✉email: y-sato4@md.tsukuba.ac.jp

The genetic determinants of mammalian immunity are commonly studied using genetically modified mice. Although we can generate transgenic mice for the phenotypic and functional analysis of immune cells in vivo, the process of generating and characterizing these mice models take several years and requires highly technical and expensive methodologies[1–4]. This hinders research progress in immunology and hematology. A quicker alternative to the generation of transgenic mice is the use of BM chimeras, which are routinely used in immunological research[5–7] and involve the engraftment and immune system-reconstitution of donor hematopoietic stem/progenitor cells (HSPCs) in recipient mice. When combined with ex vivo genetic manipulation (e.g., lentiviral transduction[8]), BM chimeras can be a useful approach to investigate the determinants of hematopoietic and immune system development and function[9].

While relatively easy and inexpensive, BM chimera assays currently have several limitations. These include the need to treat the recipient mice using irradiation or chemotherapy to allow for donor cell engraftment within the BM[10,11]. Such conditioning damages endogenous hematopoietic tissues such as the BM microenvironment and can irreversibly alter steady-state hematopoiesis[12–14]. Additionally, most protocols use complex multicolor fluorescence-activated cell sorting (FACS) to isolate donor HSPCs for transduction and transplantation, requiring expensive equipment and technical expertise.

We recently developed a polyvinyl alcohol (PVA)-based media that could expand FACS-purified $CD150^+CD34^{-/lo}c$-$Kit^+Sca1^+$ Lineage$^-$ hematopoietic stem cells (HSCs) for over a month[15,16]. Undifferentiated HSCs were stimulated to grow in these cultures using a combination of recombinant stem cell factor (SCF) and thrombopoietin (TPO) cytokines, while downstream cell types were poorly supported. Here, we describe the application of this media to enrich HSPCs from BM without the need for FACS. These cultures enriched and expanded HSPCs can be genetically modified ex vivo and robustly engraft in non-conditioned immunocompetent recipients, thereby providing a useful and practical experimental approach for the fields of immunology and hematology.

## Results

**Culture-based enrichment of HSPCs from c-Kit$^+$ BM cells.** Based on the selective expansion of HSCs in our recently described PVA-based media[15,16], we hypothesized that we may be able use ex vivo culture to enrich for functional HSCs from more differentiated cell types. To test this hypothesis, we collected c-Kit$^+$ BM cells from C57BL/6-CD45.1 mice (via magnetic column enrichment) and cultured them for 28 days ex vivo in PVA-based HSC medium (Fig. 1A). In these experiments, cell cultures were initiated with $3.5 \times 10^6$ cells, where Lin$^-$ and c-Kit$^+$Sca1$^+$Lineage$^-$ (KSL) HSPC fractions were at initial frequencies of ~71% and ~5.9%, respectively (meaning each culture was initiated with ~$2.5 \times 10^6$ Lin$^-$ cells including ~$2.1 \times 10^5$ KSL cells). A progressive increase in total cell number was observed throughout these 28-day cultures (Fig. 1B) and flow cytometric analysis detected a high percentage of c-Kit$^+$ cells (88% at day 28) (Fig. 1C, Supplementary Fig. 1A). This represented a 5.8-fold increase in c-Kit$^+$ cell numbers over 28 days (Fig. 1B, C). A more detailed analysis of phenotypic KSL and CD150$^+$KSL HSPC populations also identified progressive increases in the frequency of these populations during the culture (Fig. 1C, Supplementary Fig. 1A). These results supported our hypothesis that PVA-based HSC media could enrich for and support expansion of HSPCs ex vivo.

To evaluate the function of these culture-enriched HSPCs, we performed competitive BM transplantation using the congenic

C57BL/6 system, where irradiated C57BL/6-CD45.2 recipient mice are reconstituted with C57BL/6-CD45.1 donor cells and C57BL/6-CD45.1/CD45.2 competitor BM cells, allowing for donor reconstitution to be tracked and quantified using allele-specific CD45 antibodies (Supplementary Fig. 1B). In this study, $1 \times 10^6$ expanded cultured cells at day 28 (approximately ~1/25 of the culture, equivalent to ~$1 \times 10^5$ Lin$^-$ cells containing ~$8.4 \times 10^3$ KSL cells at day 0) or $1 \times 10^6$ fresh BM cells derived from C57BL/6-CD45.1 BM were transplanted against $1 \times 10^6$ C57BL/6-CD45.1/CD45.2 BM competitors. Expanded cells displayed stable high-level and multilineage peripheral blood (PB) reconstitution in primary recipients, similar to fresh BM cells (average of 73% vs 61% donor chimerism at 16-weeks, respectively) (Fig. 1D). In addition, there were no significant differences in multilineage differentiation within PB myeloid cells (Mac-1$^+$/Gr-1$^+$), T cells (CD4$^+$/CD8a$^+$), and B cells (B220$^+$) as compared with the fresh BM cells (Fig. 1E). Endpoint BM analysis of these primary recipients also revealed no significant differences in donor chimerism within the BM Lin$^-$, KSL, or CD34$^-$KSL population at 16 weeks post-transplant between 28-day expanded and fresh donor cells (Fig. 1F, Supplementary Fig. 1C).

To further confirm long-term (LT)-HSC function within the expanded cells, we performed secondary transplantation assays. The expanded cells from c-Kit$^+$ BM displayed higher engraftment in secondary recipients than the fresh controls throughout the 16-week assay (average 77% vs 55% donor chimerism at 16 weeks, respectively) (Fig. 1G). Finally, BM analysis of these secondary recipients after 16-weeks confirmed robust contributions of the expanded cells to the phenotypic KSL HSPC and CD34$^-$KSL HSC populations (Fig. 1H). Together, these results suggested that we could selectively expand functional HSCs ex vivo using PVA-based media.

To further assess the expansion of engraftable HSCs in these c-Kit$^+$ BM cultures, we performed limiting dilution transplantation assays comparing fresh c-Kit$^+$ cells and expanded c-Kit$^+$ cells. Doses of $1 \times 10^3$, $1 \times 10^4$, and $1 \times 10^5$ cells were transplanted into lethally irradiated C57BL/6-CD45.2 recipient mice against $1 \times 10^6$ competitor BM cells. At 12 weeks post-transplantation, average chimerism of the cultured cells was 1.3, 28, and 83%, respectively, while chimerism of fresh cells averaged 0.6, 8.7, and 67%, respectively (Fig. 1I). Using extreme limiting dilution analysis[17] (with a 1% donor chimerism threshold cutoff), we estimated the engraftable HSC frequency in the expanded cells as 1:2164 cells whereas fresh c-Kit$^+$ cells had an HSC frequency of 1:3971. These data suggest an ~8-fold expansion in HSPC potency can be achieved via this approach. We therefore conclude that the expansion culture from c-Kit$^+$ BM cells is a practical approach for efficiently generating functional HSPCs without either expensive equipment or time-intensive methods.

Given the ability of PVA-based media to enrich for HSCs in culture, we also evaluated whole (unfractionated) BM in our ex vivo culture system, with cultures initiated with $2 \times 10^7$ BM cells (Supplementary Fig. 2A). Within the starting whole BM, the Lin$^-$ and KSL fractions were at initial frequencies of ~2.3% and ~0.1%, respectively (meaning each cell culture was initiated with approximately $4.7 \times 10^5$ Lin$^-$ cells including $2.4 \times 10^4$ KSL cells). Although c-Kit$^+$ cells were enriched for over the 28-day cultures, the purity of phenotypic KSL and CD150$^+$KSL remained lower than in the c-Kit$^+$ BM-derived cultures (Supplementary Fig. 2B–D). Primary and secondary transplantation assays were also performed. In these experiments, cell cultures contained ~$1 \times 10^7$ cell by day 28, and primary recipients received $1 \times 10^6$ of these cultured cells (approximately ~1/10 of the culture, equivalent to ~$4.7 \times 10^4$ Lin$^-$ cells with ~$2.4 \times 10^3$ KSL cells at day 0). The functional assays confirmed that hematopoietic progenitors and short-term HSCs expanded in whole BM cell

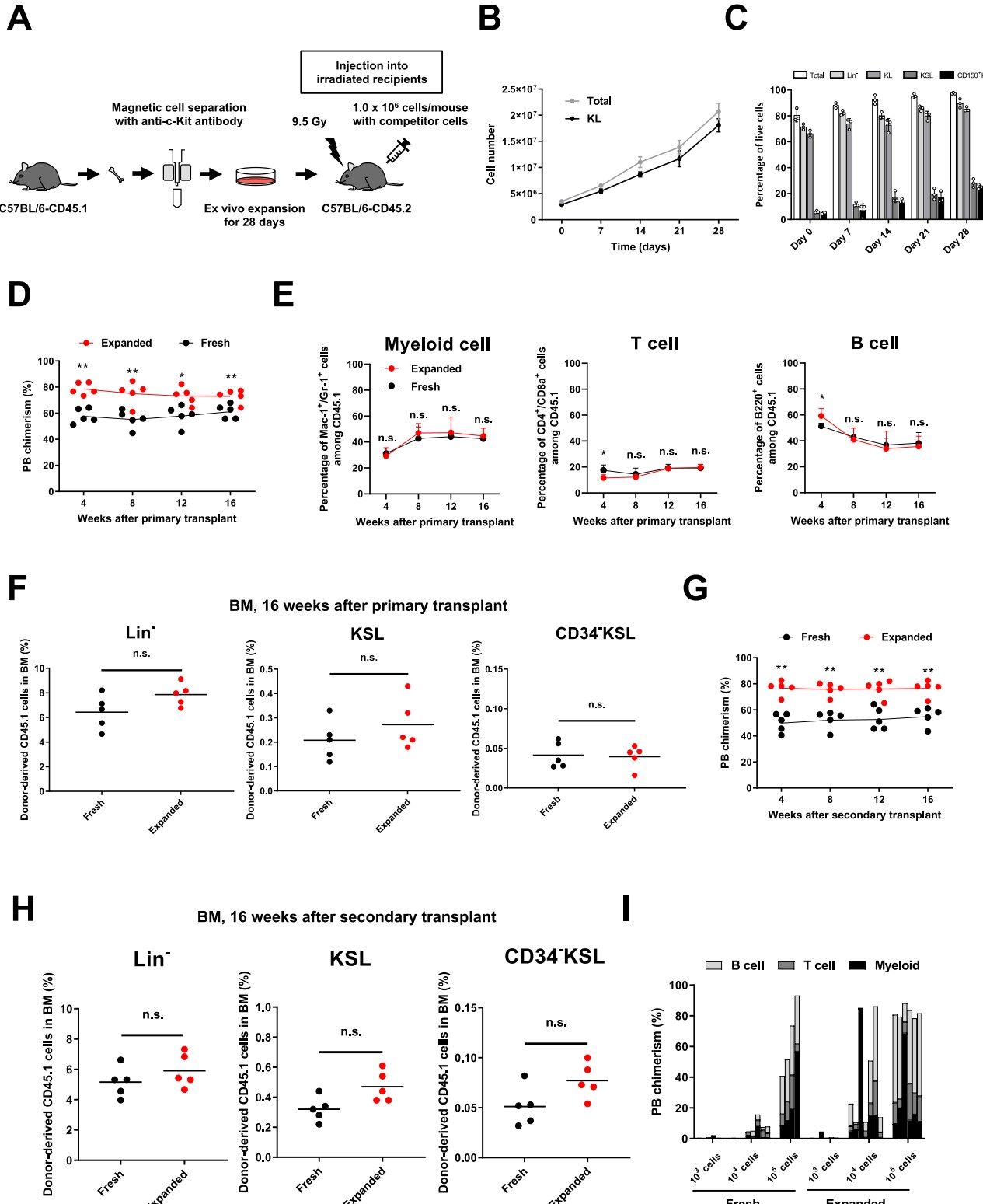

cultures (Fig. 2A, B), but low BM chimerism in primary recipients and reduced engraftment in secondary recipients suggested that LT-HSCs were poorly supported (Fig. 2C, D, Supplementary Fig. 2E).

Given that total live cell numbers rapidly dropped in the first 7 days, over 80% to ~$3.8 \times 10^6$ cells (Supplementary Fig. 2B), we hypothesized that toxicities from cell death may affect HSC activity in whole BM cultures. In order to evaluate this further, we

compared whole BM and c-Kit$^+$ BM cell cultures by performing Annexin V apoptosis assays and evaluating the expression of senescence-related genes ($p16^{Ink4a}$, $p19^{Arf}$, and $Trp53$). The frequency of apoptotic cells was significantly higher in whole BM cell cultures (Fig. 2E), and expression of $p16^{Ink4a}$ and $p19^{Arf}$ was also higher within both bulk culture cells as well as phenotypic KSL cells in the whole BM cultures (Fig. 2F). Additionally, expression of $Trp53$ was higher in the whole BM-

**Fig. 1 Ex vivo enrichment and expansion of hematopoietic stem/progenitor cell (HSPCs) from c-Kit+ BM cells. A** Experimental schematic: $3.5 \times 10^6$ c-Kit+ BM cells were plated in PVA-based HSC media and cultured for 28 days. Cell cultures were analyzed by cell counting and flow cytometry every 7 days. After 28 days, $1 \times 10^6$ cultured cells were transplanted into irradiated C57BL/6-CD45.2 recipients ($n = 5$ mice per group) alongside $1 \times 10^6$ C57BL/6-CD45.1/CD45.2 whole BM competitor cells. Donor PB chimerism was quantified over 16 weeks, at which point BM chimerism and secondary transplantation assays were performed. **B** Total and KL cell numbers during the ex vivo culture. Mean ± SD of independent triplicate cultures. **C** Percentage of Lin−, KL, KSL, and CD150+KSL phenotypic cell populations during the ex vivo culture. Mean ± SD of independent triplicate cultures. **D** Percentage of donor-derived CD45.1+ cells in primary transplant recipients over time ($n = 5$ mice per group). **$p = 0.0002$ at week 4, **$p = 0.0038$ at week 8, *$p = 0.0111$ at week 12, **$p = 0.0082$ at week 16. **E** Percentage of Mac-1+/Gr-1+ (myeloid cells), CD4+/CD8a+ (T cells), and B220+ (B cells) among donor-derived CD45.1+ cells in primary transplant recipients over time. Mean ± SD from 5 primary recipients ($n = 5$ mice per group). *$p = 0.0262$ at week 4 in T cell, *$p = 0.0208$ at week 4 in B cell. **F** Percentage of Lin−, KSL, and CD34−KSL cells in BM cells among donor-derived CD45.1+ cells at 16 weeks after primary transplantation ($n = 5$ mice per group). **G** Percentage of donor-derived CD45.1+ cells over time in secondary transplant recipients ($n = 5$ mice per group).) **$p = 0.0002$ at week 4, **$p = 0.0003$ at week 8, **$p = 0.0013$ at week 12, **$p = 0.0008$ at week 16. **H** Percentage of Lin−, KSL, and CD34−KSL cells in BM cells among donor-derived CD45.1+ cells at 16 weeks in secondary recipients. Mean ± SD from 5 primary recipients ($n = 5$ mice per group). **I** Limiting dilution assay in conditioned recipients. $1 \times 10^3$, $1 \times 10^4$, $1 \times 10^5$ expanded c-Kit+ BM cells at day 28 of culture were transplanted into lethally irradiated C57BL/6-CD45.2 recipient mice with $1 \times 10^6$ C57BL/6-CD45.1/CD45.2 whole BM competitor cells. Donor PB chimerism at week 12 in primary recipient mice ($n = 8$ mice per group) with percentages of Mac-1+/Gr-1+ (myeloid cells), CD4+/CD8a+ (T cells), and B220+ (B cells) displayed. Statistical significance was calculated using an unpaired two-tailed $t$ test, *$p < 0.05$, **$p < 0.01$; n.s. not significant.

derived bulk cell cultures (Fig. 2F). Together, these results suggested that cellular stress may be negative regulator of HSC activity ex vivo.

**Cultured HSPCs efficiently engraft non-conditioned mice.** Generation of mice chimeric BM without conditioning represents an important approach to study immune system development and function without the toxicities associated with irradiation or chemotherapy. Unfortunately, the routine use of approach has been essentially unfeasible due to the large numbers of HSPCs required for engraftment; donor HSPCs engraft very inefficiently without recipient conditioning[18]. In our previous study, the HSPCs expanded from FACS-purified CD150+CD34−KSL cells were able to engraft in immunocompetent autologous mice without irradiation or other conditioning[15].

We therefore sought to evaluate the capacity of our culture-enriched HSPCs from c-Kit+ BM cells to engraft and reconstitute non-conditioned recipients by transplanting $1 \times 10^6$ culture-enriched cells (from C57BL/6-CD45.1 c-Kit+ BM) into C57BL/6-CD45.1/CD45.2 recipient mice (Supplementary Fig. 3A). Donor engraftment and lineage contribution was track by PB chimerism and compared with endogenous CD45.1/CD45.2 lineage contribution (Fig. 3A). Robust PB chimerism from culture-enriched HSPCs was observed in non-conditioned recipients, averaging 30–35% between 4 and 16-weeks post-transplantation (Fig. 3B). Consistent with immune lineage differentiation kinetics, 4-week engraftment was dominated by B cells, while T cells took longer to be generated (Fig. 3C). Additionally, when compared to endogenous CD45.1/CD45.2 hematopoiesis, no significant differences were observed within the BM Lin−, KSL, or CD34−KSL populations at 16 weeks post-transplantation (Supplementary Fig. 3B). These results suggested that culture-enriched HSPCs could robustly engraft in the BM of non-conditioned recipients and contribute to immune system formation.

To further validate the function of these engrafted HSPCs in primary recipients, secondary transplantation analysis was performed into C57BL/6-CD45.2 recipients. Stable donor chimerism was observed for the entire 16-week analysis, with mean PB chimerism at 30–31% (Supplementary Fig. 3C). Additionally, consistent donor chimerism was also observed within the BM Lin−, KSL, and CD34−KSL cell fractions (Supplementary Fig. 3D). These results confirmed that non-conditioned transplantation of expanded cells resulted in the engraftment of functional LT-HSCs.

Finally, to determine the potential of ex vivo expanded c-Kit+ BM, we performed limiting dilution transplantation assays, comparing fresh c-Kit+ BM cells and 28-day cultured cells in non-conditioned recipients. Doses of $1 \times 10^4$, $1 \times 10^5$, $1 \times 10^6$ cells were transplanted into C57BL/6-CD45.1/CD45.2 recipients. After 12 weeks of transplantation, the average chimerism of fresh c-Kit+ cells was 0.1, 0.2, and 0.8%, respectively, while the average chimerism of cultured cells was 0.1, 2.6, and 20%, respectively (Fig. 3D). These results confirmed that culture-enriched HSPCs have high engraftment and reconstitution potential, and at least 10-fold higher engraftment potential than fresh c-Kit+ cells. It is worth noting the remarkable advantages of these non-conditioned BM chimeric mice; as this procedure does not involve radiation or chemotherapy, immune system and function can be studied under steady-state conditions, and the lack of undesirable morbidity/mortality from cytotoxic chemotherapy and irradiation-based conditioning means that recipients can survive long-term (more than a year) post-transplantation (Fig. 3E).

**Genetically engineered immune cells from cultured HSPCs.** As demonstrated above, these 28-day ex vivo cultures enriched and expanded HSPCs with long-term reconstitution activity and immune cell repopulation potential in vivo. We therefore evaluated the possibility of genetic manipulation and ex vivo selection using lentiviral vectors in this culture platform. As proof-of-concept, we developed a puromycin-selectable doxycycline (DOX) inducible vector system for inducible transgene expression (Supplementary Fig. 4A), and evaluated expression of enhanced green fluorescent protein (EGFP), AE9a (an AML1/ETO isoform)[19], and BCR-ABL[20].

Lentiviral transduction was first evaluated ex vivo by EGFP expression and drug selection efficiencies (Fig. 4A). In these experiments, c-Kit+ BM cells were seeded in PVA-based HSC medium, transduced lentivirus particles on day 2, and then treated with puromycin for 48 h at day 4. After puromycin selection, replace with fresh medium without puromycin. The 28-day cultures maintained a high percentage of c-Kit positive cells (average 92%) (Supplementary Fig. 4B) and displayed a 4.8-fold increase in total CD150+KSL cell numbers (Supplementary Fig. 4C). After a total of 28 days culture, the cells were split into media with or without DOX for 48 h, and then analyzed by flow cytometry and RT-PCR. In terms of the frequencies of phenotypic Lin−, c-Kit+Lineage− (KL) and KSL cell fractions, transduced cells showed no difference compared to non-

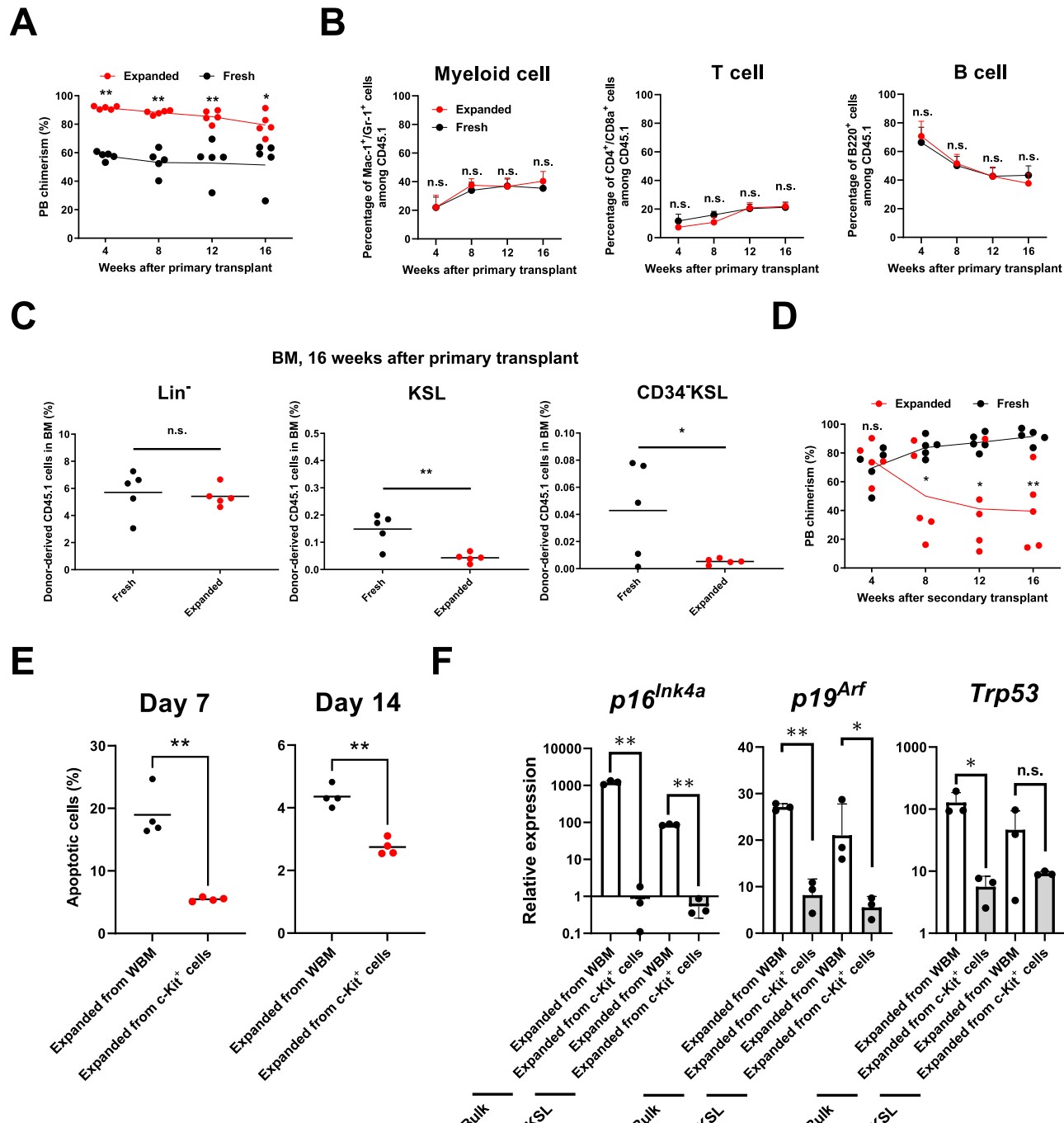

**Fig. 2 Ex vivo enrichment of HSPCs from whole BM cells. A** Percentage of donor-derived CD45.1⁺ cells in primary transplant recipients over time ($n = 5$ mice per group) following competitive transplantation of $1 \times 10^6$ fresh or 28-day cultured whole BM cells (same transplantation protocol as detailed in Fig. 1A). **$p < 0.0001$ at week 4, **$p < 0.0001$ at week 8, **$p = 0.0013$ at week 12, *$p = 0.0114$ at week 16. **B** Percentage of Mac-1⁺/Gr-1⁺ (myeloid cells), CD4⁺/CD8a⁺ (T cells), and B220⁺ (B cells) among donor-derived CD45.1⁺ cells in primary transplant recipients over time for the mice described in **A**. Mean ± SD from 5 primary recipients ($n = 5$ mice per group). **C** Percentage of CD45.1⁺ BM Lin⁻, KSL, and CD34⁻KSL cells at 16 weeks post-transplantation for the mice described in **A** ($n = 5$ mice per group). **$p = 0.0043$ in KSL, *$p = 0.0142$ in CD34⁻KSL. **D** Percentage of donor-derived CD45.1⁺ cells over time following secondary transplantation using the mice described in **A** ($n = 5$ mice per group). $p = 0.6265$ at week 4, *$p = 0.0459$ at week 8, *$p = 0.0107$ at week 12, **$p = 0.0024$ at week 16. **E** Percentages of Annexin V⁺/Propidium iodide (PI)⁺ from day 7 and day 14 cultured whole BM or c-Kit⁺ BM cells. Mean ± SD of 4 independent cultures **$p = 0.0004$ at Day7, **$p = 0.0003$ at Day 14. **F** Evaluation of gene $p16^{Ink4a}$, $p19^{Arf}$ and $Trp53$ expression in 7-day cultured whole BM or c-Kit⁺ BM cells (RNA isolated from bulk and phenotypic KSL cells). Mean of three independent cultures, with gene expression normalized to *Gapdh* expression. Error bar denote SD. $p16^{Ink4a}$: **$p = 0.0001$ (bulk cells expanded whole BM vs c-Kit⁺ cells), **$p < 0.0001$ (KSL expanded from whole BM vs c-Kit⁺ cells), $p19^{Arf}$:**$p = 0.0008$ (bulk cells expanded from whole BM vs c-Kit⁺ cells), *$p = 0.0203$ (KSL expanded from whole BM vs c-Kit⁺ cells), $Trp53$: *$p = 0.0218$ (bulk cell expanded from whole BM vs c-Kit⁺ cells), $p = 0.2424$ (KSL expanded whole BM vs c-Kit⁺ cells). Statistical significance was calculated using an unpaired two-tailed *t*-test, *$p < 0.05$, **$p < 0.01$; n.s. not significant.

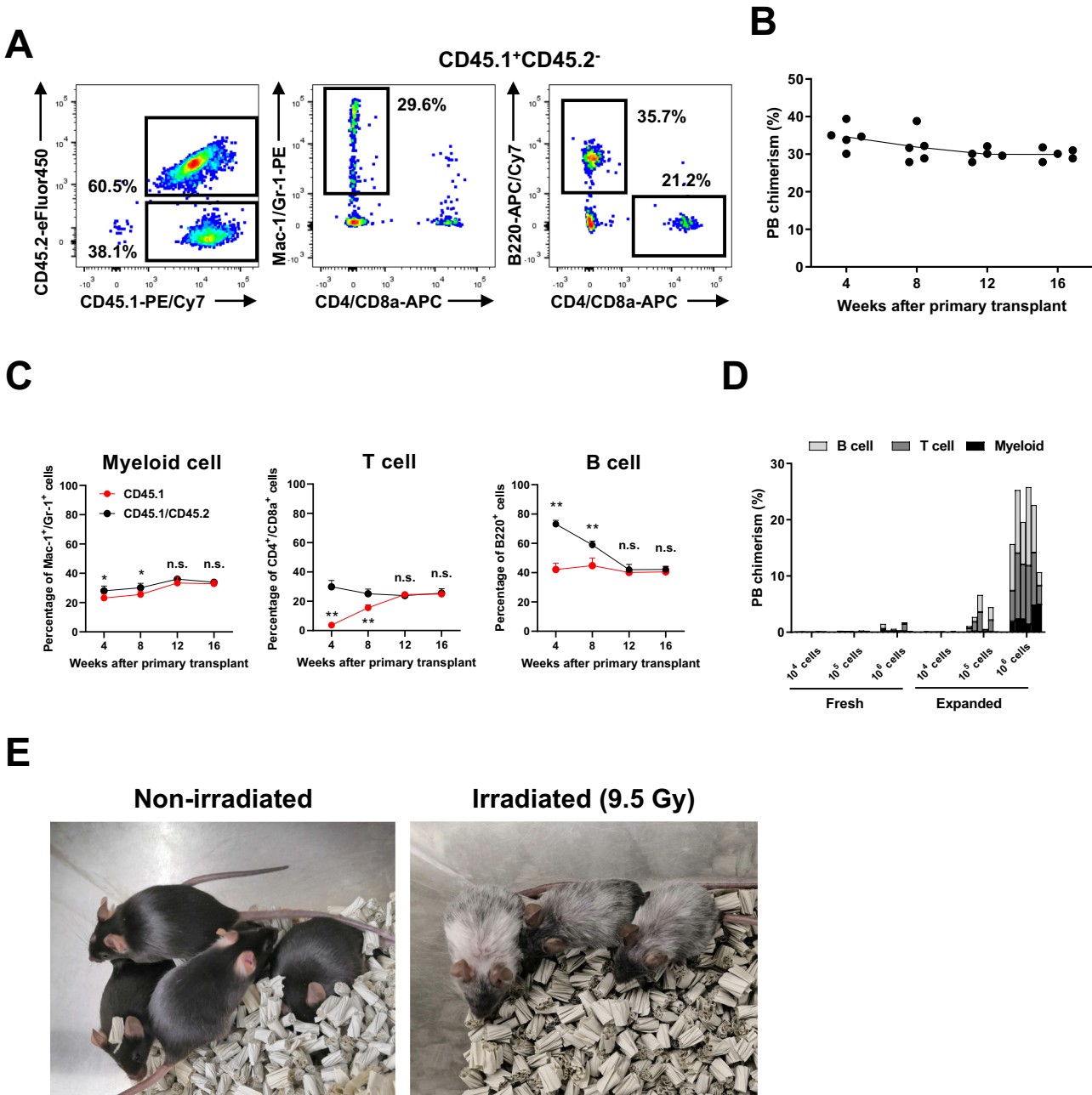

**Fig. 3 Generation of irradiation-free BM chimeric mice. A** Gating strategy used to detect donor-derived CD45.1⁺ PB chimerism in non-conditioned C57BL/6-CD45.1/CD45.2 recipients. **B** Percentage of donor-derived CD45.1⁺ cells after primary transplant of $1 \times 10^6$ 28-day cultured c-Kit⁺ BM cells ($n = 5$ mice). **C** Percentage of Mac-1⁺/Gr-1⁺ (myeloid cells), CD4⁺/CD8a⁺ (T cells), and B220⁺ (B cells) among donor- and recipient-derived CD45.1⁺ cells in primary transplant recipients described in **A**. Mean ± SD from 5 primary recipients ($n = 5$ mice per group). *$p = 0.0327$ at week 4 and *$p = 0.0225$ at week 8 in myeloid cell, **$p < 0.0001$ at week 4 and **$p = 0.0005$ at week 8 in T cell, **$p < 0.0001$ at week 4 and **$p = 0.0007$ at week 8 in B cell. **D** Limiting dilution assay in non-conditioned recipients. $1 \times 10^4$, $1 \times 10^5$, $1 \times 10^6$ fresh or 28-day expanded c-Kit⁺ BM cells were transplanted into non-conditioned C57BL/6-CD45.1/CD45.2 recipient mice. Donor PB chimerism at week 12 in primary recipient mice ($n = 6$ mice per group) with percentages of Mac-1⁺/Gr-1⁺ (myeloid cells), CD4⁺/CD8a⁺ (T cells), and B220⁺ (B cells) displayed. **E** Gross appearances of representative recipient mice at 16 weeks after primary transplant. Recipient mice transplanted without irradiation (left), and mice transplanted with doses of 9.5 Gy irradiation (right). Statistical significance was calculated using an unpaired two-tailed *t*-test, *$p < 0.05$, **$p < 0.01$; n.s. not significant.

transduced (mock) cells (Fig. 4B). Quantitative PCR confirmed higher *EGFP* expression in the presence of DOX (Fig. 4C), which was also confirmed by flow cytometry (Fig. 4D, E). As compared to cells without puromycin treatment, puromycin selection significantly increased the frequency of EGFP⁺ cells (average of ~23% vs 98% within the KSL cell population) (Fig. 4D, E, Supplementary Fig. 4D).

Having validated the stable transduction of culture-enriched HSPCs, we next evaluated the reconstitution and EGFP expression in vivo (Fig. 4A). We injected $1 \times 10^6$ cells into each non-conditioned immunocompetent and tracked donor chimerism over 16 weeks; stable multilineage PB chimerism at 22–26% was observed during this time (Fig. 4F). To evaluate inducible transgene expression in vivo, recipient mice were administrated

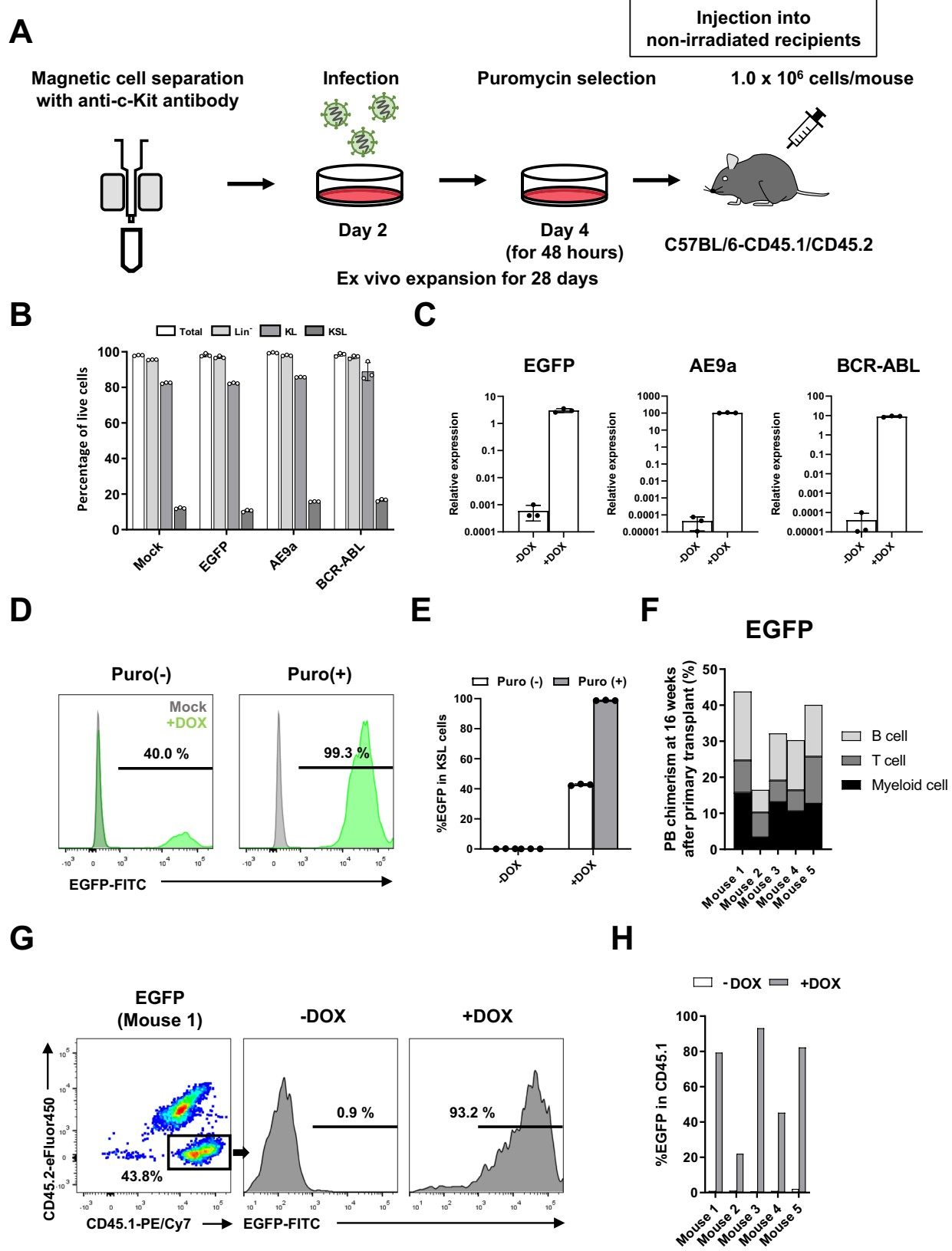

with DOX via the drinking water ad libitum at 16 weeks post-transplantation[21,22]. After DOX administration, >90% of CD45.1[+] donor-derived cells displayed EGFP expression and large increases in EGFP gene expression were observed (Fig. 4G, H, Supplementary Fig. 4E). By contrast, minimal leaky EGFP expression (1–2%) was observed prior to DOX administration (Fig. 4G, H, Supplementary Fig. 4E).

These experiments were also repeated with AE9a and BCR-ABL transgenes. Inducible expression of AE9a and BCR-ABL expression was confirmed in ex vivo cell cultures and in CD45.1[+]

**Fig. 4 Conditional transgene expression in non-conditioned BM chimeric mice. A** Experimental schematic for non-conditioned transplantation of culture-enriched HSPC cultures derived from c-Kit[+] BM cells following lentiviral transduction and puromycin selection. Cell cultures were initiated with $3.5 \times 10^6$ c-Kit[+] BM cells. Lentiviral transduction was performed at day 2, puromycin treatment performed at day 4 for 48 h, DOX treatment performed at day 28 for 48 h, and flow cytometric analysis on day 30. $1 \times 10^6$ day-28 cells were injected into non-conditioned C57BL/6-CD45.1/CD45.2 recipient mice. **B** Percentage of phenotypic Lin[−], KL and KSL cell populations at culture day 28, as described in **A**. Mean ± SD of independent triplicate cultures. **C** Quantitative-PCR for expression of *EGFP*, *AE9a*, and *BCR-ABL* in cells from DOX on or off HSPC cultures. Mean ± SD of independent triplicate cultures. **D** Representative flow cytometric histograms for EGFP expression in HSPC cultures with or without puromycin selection. **E** Percentages of EGFP[+] cells in KSL cells with or without puromycin selection. Mean ± SD of independent triplicate cultures. **F** PB chimerism and multilineage differentiation of CD45.1[+] cells at 16 weeks post transplant ($n = 5$ mice), for the non-conditioned recipients described in **A**. **G** Representative flow cytometric plots of CD45.1[+] cells and EGFP expression in non-conditioned recipients (described in **F**) before and after DOX administration at 16-weeks post-transplantation. **H** Percentage of EGFP in CD45.1[+] PB cells in individual mice (described in **F**) before and after DOX administration at 16-weeks post-transplantation ($n = 5$ mice).

PB cells following non-conditioned transplantation (Fig. 4B, C, Supplementary Fig. 4F–H). Finally, we evaluated the consequences of lentiviral transduction on the engraftment potential of culture-enriched HSPCs using the EGFP-transduced HSPC recipient cohort. By flow cytometry, no significant differences were observed in the frequency of phenotypic Lin[−], KSL, and CD34[−]KSL BM cell populations (Supplementary Fig. 4I). Additionally, stable long-term PB and BM chimerism was observed in secondary transplantation assays (Supplementary Fig. 4J–K). Together, these results confirm that our HSPC-enrichment cultures can be used to efficiently produce genetically engineered LT-HSCs and immune cells in vivo.

## Discussion

While hematopoietic and immune system reconstituting HSCs remain the most well-characterized somatic stem cell population, the isolation of HSCs usually involves complex and expensive methodologies[23–25]. Here, we have demonstrated that functional HSCs can be enriched by simply culturing magnetic column-enriched c-Kit[+] BM cells in PVA-based HSC medium[15]. This culture-based enrichment retains functional LT-HSCs for at least 28 days ex vivo. As demonstrated here, these cell culture-enriched HSPCs are also amenable to viral transduction and further selection prior to transplantation. This approach therefore has broad and important applications in the study of immune system function. Furthermore, the large numbers of HSPCs that can be generated by this approach allow for the engraftment of genetically modified HSPCs in non-conditioned immunocompetent mice, affording studies to be undertaken without tissue toxicities associated with chemotherapy or irradiation conditioning.

In conclusion, we have developed a simple and useful approach for generating irradiation-free BM chimeric mice for the study of steady-state hematopoiesis and immune system development. The approach improves experimental animal welfare and does not require any specialized equipment (e.g., multicolor FACS). We therefore expect that this platform has numerous applications for the investigation of hematopoietic system development and immune response in health and disease.

## Methods

**Mice.** C57BL/6-CD45.1 (CD45.1) and C57BL/6-CD45.1/CD45.2 (CD45.1/CD45.2) mice were purchased from Sankyo-Laboratory Service (Tsukuba, Japan), C57BL/6-CD45.2 (CD45.2) mice were purchased from Japan SLC (Shizuoka, Japan), respectively. 8 to 12-week-old male mice were used as donors and recipients. All mice were housed in specific-pathogen-free conditions with free access to food and water. All animal protocols were approved by the Animal Care and Use Committee of the Institute of Medical Science, the University of Tokyo. The environment in the mouse chamber is a temperature of 23–25 °C, humidity of about 50%, and a light period of 12 h each.

**PVA-based serum-free culture.** C57BL/6-CD45.1 mouse BM cells, unfractionated or following magnetic c-Kit[+] cell enrichment, were cultured using Ham's F-12 medium (Wako), supplemented with 0.1% PVA (Sigma, Cat# P8136), 1% Insulin-Transferrin-Selenium-Ethanolamine (ITS-X) (100X) (Thermo Fisher Scientific), 1%

Penicillin-Streptomycin-L-Glutamine Solution (100X) (Wako), N-2-hydroxyethylpiperazine-N-2-ethane sulfonic acid (HEPES) (10 mM; Gibco), mouse TPO (100 ng/ml; PeproTech), and mouse SCF (10 ng/ml; PeproTech) for 28 days, incubated at 37 °C in a humidified 5% $CO_2$ incubator. Medium was changed every other day[15,16]. Magnetic cell separation was performed using anti-mouse c-Kit MicroBeads (Miltenyi Biotech, Cat# 130-091-224) according to the manufacturer's instructions. Cell culture using either commercially available pre-packaged HemEx-Type9A (NIPRO) or in-house prepared medium supplemented with 100 ng/mL mouse TPO and 10 ng/mL mouse SCF. Unfractionated whole BM cells were seeded at $2 \times 10^6$/mL onto 100 mm dish in 10 mL culture medium (day 0–14) or 60 mm dish in 4 mL culture medium (day 15–28). Magnetic column-enriched c-Kit[+] BM cells were seeded at $1 \times 10^6$/well onto 48-well plates in 1 mL culture medium. For long-term cultures, complete medium changes were made every 2 days and cell cultures were passaged at a ratio of 1:2-3 when cells exhibited 80–90% confluency.

**Cell counting and sample preparation for flow cytometry.** Cell counting was performed using an automated cell counter (Countess II Automated Cell Counter, Invitrogen). PB samples were collected from the retro-orbital venous plexus into capillary tubes filled with powdered EDTA. BM samples were obtained by flushing the tibia, femur, and pelvic bones with sterile 1X phosphate-buffered saline. PB samples were stained with PE-conjugated anti-Gr-1, PE-conjugated anti-Mac-1, APC-conjugated anti-CD4, APC-conjugated anti-CD8a, APC/Cy7-conjugated anti-B220, PE/Cy7-conjugated anti-CD45.1, and eFluor450-conjugated anti-CD45.2 antibodies. BM samples were stained with PE/Cy7-conjugated anti-CD45.1, eFluor450-conjugated anti-CD45.2, PE-conjugated anti-Sca-1, APC-conjugated anti-c-Kit, APC/eFluor780-conjugated anti-CD4, APC/eFluor780-conjugated anti-CD8, APC/eFluor780-conjugated anti-Mac-1, APC/eFluor780-conjugated anti-Gr-1, APC/eFluor780-conjugated anti-B220, APC/eFluor780-conjugated anti-Ter119, APC/eFluor780-conjugated anti-D127/IL-7Rα, and FITC-conjugated anti-CD34 or PE/Cy5-conjugated anti-CD34antibodies. Cultured BM samples were stained with PE-conjugated anti-CD150, APC-conjugated anti-c-Kit, PE/Cy7-conjugated anti-Sca-1, APC/eFluor780-conjugated anti-CD4, APC/eFluor780-conjugated anti-CD8, APC/eFluor780-conjugated anti-Mac-1, APC/eFluor780-conjugated anti-Gr-1, APC/eFluor780-conjugated anti-B220, APC/eFluor780-conjugated anti-Ter119, and APC/eFluor780-conjugated anti-D127/IL-7Rαantibodies. See Supplementary Table 1 for antibody details. Data were acquired on a FACSAriaIII or FACSVerse (BD) and analyzed with FlowJo (v10.5.3) Software (FlowJo, LLC).

**Competitive transplantation assays.** Fresh or cultured cells (at indicated cell doses) were transplanted via single intravenous injection into irradiated (9.5 Gy) male C57BL/6-CD45.2 recipient mice along with $1 \times 10^6$ male C57BL/6-CD45.1/ CD45.2 whole BM competitor cells. PB analysis was performed every 4 weeks for 16 weeks. Limiting dilution analysis was performed using ELDA software based on a 1% PB multilineage chimerism as the threshold for positive engraftment.

**Non-conditioned transplantation assays.** Fresh or cultured cells (at indicated cell doses) were transplanted via single intravenous   injection into non-irradiated male C57BL/6-CD45.1/CD45.2 recipient mice. PB analysis was performed every 4 weeks for 16 weeks.

**Secondary BM transplantation assays.** Secondary BM transplantation assays were performed after 16 weeks by transferring $1 \times 10^6$ whole BM cells from the primary recipient mice into irradiated (9.5 Gy) male C57BL/6-CD45.2 mice by intravenous injection.

**Vector construct and lentivirus production.** The DOX-inducible lentiviral vector was based on an all-in-one inducible lentiviral vector (Ai-LV)[26] from Dr. T. Yamaguchi (The University of Tokyo), and was cloned to carry a Tet-responsive promoter driving EGFP, AE9a or BCR-ABL and a downstream rtTA gene and puromycin gene driven by a human elongation factor-1 alpha (EF-1a) promoter. MigR1-AE9a and P210 pcDNA3 plasmids were used as templates for AE9a and BCR-ABL transgenes. Viral supernatant was generated from a lentiviral vector

transfected into Lenti-X 293 T Cells using a polyethyenimine (PEI) transfection protocol. For each 10 cm culture dish, use the lentiviral transfer vector plasmid, packaging plasmid (psPAX2), and envelope plasmid (pMD2G) DNA (6:3:1.5 µg, respectively) in 500 µL of 10 mM HEPES/150 mM NaCl (pH 7.05). Add 42 ul of 1 ug/uL PEI MAX (Polysciences Inc) to the diluted DNA. DNA/PEI mixtures were incubated at room temperature for ≥10 min. Lenti-X 293 T Cells were trypsinized, washed twice with 1X phosphate buffered saline, and resuspended ($2 \times 10^6$ cells/ml) in serum-free Opti-MEM (Invitrogen Life Technologies, Inc.). The DNA/PEI mixture was added to 7.5 ml suspended cells, and immediately plated onto a 10 cm dish and incubated at 37 °C in a 5% $CO_2$ humidified atmosphere. The supernatants were collected and concentrated by using Lenti-X concentrator (Takara Bio Inc.), and stored at −80 °C. MigR1-AE9a was a gift from Dong-Er Zhang (Addgene plasmid #12433)[19]. P210 pcDNA3 was a gift from Warren Pear (Addgene plasmid #27481)[20]. psPAX2 and pMD2.G were a gift from Didier Trono (Addgene plasmid #12260 and #12259).

**Viral transfection and puromycin selection**. Mouse c-Kit$^+$ BM cells were transduced in the RetroNectin-bound virus (RBV) method according to the manufacturer's instructions (RetroNectin, Takara Bio Inc.). In brief, 48-well tissue culture plates were coated with RetroNectin Reagent at 20 µg/cm² prior to cell seeding. After 2 days in culture, cells were transduced by spinoculation (2 h, 2000 g, 32 °C) at a concentration of 5000–100,000 cells/mL at multiplicities of infection of 500, and subsequently selected by puromycin for 48 h at a final concentration of 1 ug/mL from day 4 of culture. Where indicated, EGFP expression was induced with 1 µg/ml DOX (Clontech) for 48 h. Induced cells were analyzed for EGFP expression by flow cytometry. Non-transduced cells and transduced cells without puromycin selection were used as a control.

**Administration of DOX to mice**. Non-conditioned transplant male C57BL/6-CD45.1/CD45.2 mice at 16 weeks post-transplantation were administered DOX via the drinking water (2 mg/mL DOX + 1% sucrose) ad libitum for 48 h.

**Apoptosis assays**. Expanded whole BM or c-Kit$^+$ BM cells at day 7 and day 14 were stained ($1 \times 10^6$ cells per sample) with 5 µl of Brilliant Violet 421 Annexin V (Biolegend) and 10 µl of 0.5 mg/ml of propidium iodide (Biolegend)[27]. Staining patterns were collected using a FACSAriaIII or FACSVerse (BD) and analyzed with FlowJo (v10.5.3) Software (FlowJo, LLC).

**Gene expression analysis**. RNA was isolated using the NucleoSpin® RNA Plus XS (MACHERY-NAGEL) or NucleoSpin® RNA Plus (MACHERY-NAGEL). RNA was reverse-transcribed using the PrimeScript™ RT Master Mix (Perfect Real Time). For *p16ink4a p19Arf*, *Trp53* quantitative PCR was performed on a Quantstudio7 (Applied Biosystems) using THUNDERBIRD® SYBR qPCR Mix (Takara Bio Inc.) and the primer sets described in Supplementary Table 2[15,19,28]. Gene expression was normalized relative to *Gapdh* expression.

**Statistical analysis**. Statistical significance was calculated using an unpaired Student's *t* test (two-tailed), *$p < 0.05$, **$p < 0.01$. Results are shown as arithmetic means ± SD. GraphPad Prism 8 was used for statistical analysis.

**Reporting summary**. Further information on research design is available in the Nature Research Reporting Summary linked to this article.

## Data availability

Data supporting the findings of this work are available within the paper and its Supplementary Information files. A reporting summary for this Article is available as a Supplementary Information file. The datasets and materials generated and analyzed during the current study are available from the corresponding author upon request. The source data underlying Figs. 1, 2, 3, and 4, as well as Supplementary Figs. 2, 3, and 4 are provided as a Source Data file. A detailed protocol is available at the Protocol Exchange[29]. Source data are provided with this paper.

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

## Acknowledgements

We thank S. Yamazaki, Y. Ishii, R. Ishida, and HJ. Becker for support and advice. This research was funded by JSPS KAKENHI Grant-in-Aid for Scientific Research (JP20H03707; JP20H05025; JP20K17407) and the Japan Agency for Medical Research and Development (JP18bm0404025; 21bm0704055h0002). A.C.W. acknowledges funding

support from the Leukemia and Lymphoma Society (3385-19), NIH (K99HL150218), the Edward P. Evans Foundation, and the Kay Kendall Leukaemia Fund.

## Author contributions

K.O., M.M., and S.Y. conceived and designed the study, analyzed data, and wrote the manuscript. A.I. and A.C.W. wrote and edited the manuscript.

## Competing interests

The authors declare no competing interests.
