## [Peer Review File · Nature Communications]

REVIEWER COMMENTS

Reviewer #1 (Remarks to the Author):

Ochi et al adapted the mouse HSC culture protocol recently described by Wilkinson & colleagues (Nature 2019), avoiding the tedious step of obtaining highly purified HSC by multi-parameter fluorescence-activated cell sorting. They suggest that bead-selected CD117+ cells are an appropriate starting cell population for PVA/SCF/TPO-based culture, where 1 million expanded cells (the equivalent of ~172,000 Kit+ input cells considering a 5.8 fold overall cell number expansion in culture) show somewhat higher engraftment capacity than 1 million total BM cells in competitive transplantation assays into lethally irradiated mice. Considering a frequency of CD117+ cells around 5-10% in the mouse BM, this data may be more consistent with HSC maintenance rather than substantial ex vivo expansion. On the other hand, when using total BM cells as culture input, ex vivo culture appears to expand progenitors while depleting HSC, possibly due to release of stress signals in dying mature cells during the first week of culture (it would be good to show some data in support of this hypothesis). Impressively, the authors show 30% long-term engraftment after transplanting 1 million culture-enriched CD117+ cell progeny into non-conditioned mice suggesting substantial ex vivo expansion, a result which was reproduced in a second experiment with genetically-engineered and in vitro selected cells. While novelty is somewhat limited due to the previous publication, a simplified protocol including a genetic engineering/selection step that yields up to 30% of long-term engraftment in non-conditioned mice is of significant interest to the community studying immunology & hematopoiesis. The following major points should be addressed in order to ensure robustness and reproducibility of the transplantation protocol:

- 1) The authors should indicate the "day 0 equivalent" number of KSL or Lin- cells that were transplanted in the various experiments, in order to allow for a better comparison between test-, control- and supporter cell dose. Absolute numbers of KSL or CD34-KSL cells in the mouse BM (Fig.1F, 2F) should be provided, if possible.
- 2) LT-HSC expansion from Kit+ cultures should be quantified according to state-of-the-art functional assays, namely limiting-dilution transplantation and/or clonality analysis of lentivirally-marked cells. Functional data are important, as immunophenotypes revealed in culture may not have the same significance as on fresh cells.
- 3) What is the "dose-response" relationship for non-conditioned transplants? Are there threshold levels for the number of LT-HSC units/total cells in order to achieve stable engraftment in the double digit % range? What is the mechanism of HSC exchange? Is it the stoichiometric ratio between endogenous HSC and transplanted HSC? Is the timing of physiologic HSC mobilisation & homing (and therefore, the persistence of transplanted HSC in alternative niches) relevant, or can expanded HSC displace endogenous HSC from the bone marrow niche (producing a gain-of-function)? Is ex vivo culture essential, or can a "mega-dose" of fresh KSL cells provide equivalent levels of engraftment in the non-conditioned setting?
- 4) How do the authors explain the discrepancy between the %GFP+ cells in vitro (close to 100%) and in vivo (around 30%) in Fig.4? Promoter silencing? Incomplete in vitro selection at the HSC (as opposed to the KSL) level?

Reviewer #2 (Remarks to the Author):

Ochi et al describe a novel strategy that allows enrichment of functional HSPCs without sorting, and the subsequent generation of bone marrow chimeras without host conditioning. Overall the approach may prove to be quite useful and overcome some technical hurdles associated with generation of BM chimeras. Moreover, achieving robust engraftment in non-irradiated recipients may enable new

insights about hematopoiesis and the immune system in models that previously have been assessed using lethally irradiated recipients. However, the culture system has already been published as well as the generation of bone marrow chimeras without irradiation. As such the conceptual advance is incremental. The study lacks several important controls to enable assessment of the actual efficiency of the system. The authors should further demonstrate how the system may benefit the study of hematopoiesis and immune system using models with biological effects, in addition to their GFP reporter. Finally, I have some serious concerns about the rigidity of the data analysis (see below).

Specific remarks:

Figure 1: This is mostly negative data and could be moved to the supplement

Figure 2: The degree of expansion appears quite modest compared to what was previously reported for purified HSCs. A precise quantification of the actual expansion would be necessary to understand this. The competitive transplants are difficult to understand as 1 million fresh c-kit+ cells appear to give compete equally to 1 million whole BM cells (around 50% chimerism). Would why are the c-kit+ cells dominating over the whole BM cells in the long term?

Figure 3: The level of engraftment from only 1 million expanded cells is quite remarkable. However, it would be essential to include a control with equivalent and higher numbers of non-cultured cells to understand the benefit of the culture system for non-irradiated recipients.

Figure 4:

The authors should explain why the GFP induction only works in around 30% of the donor cells despite transduced cells being selected for by puromycin.

The authors should carefully reanalyze and reexamine their data and figures. Careful examination of the FACS plots in Figure 4A shows that the second and fourth plot are must be derived from the same sample but with more events shown in the fourth plot. This may be a simple mistake but is very concerning and puts into question the rigor of the entire study and the analysis of the remainder of the data presented in the paper.

We would like to thank Reviewers for their constructive feedback on our manuscript. As detailed below, we have fully resolved all concerns about our work.

REVIEWER COMMENTS

Reviewer #1 (Remarks to the Author):

Ochi et al adapted the mouse HSC culture protocol recently described by Wilkinson & colleagues (Nature 2019), avoiding the tedious step of obtaining highly purified HSC by multi-parameter fluorescence-activated cell sorting. They suggest that bead-selected CD117+ cells are an appropriate starting cell population for PVA/SCF/TPO-based culture, where 1 million expanded cells (the equivalent of ~172,000 Kit+ input cells considering a 5.8 fold overall cell number expansion in culture) show somewhat higher engraftment capacity than 1 million total BM cells in competitive transplantation assays into lethally irradiated mice. Considering a frequency of CD117+ cells around 5-10% in the mouse BM, this data may be more consistent with HSC maintenance rather than substantial ex vivo expansion. On the other hand, when using total BM cells as culture input, ex vivo culture appears to expand progenitors while depleting HSC, possibly due to release of stress signals in dying mature cells during the first week of culture (it would be good to show some data in support of this hypothesis).

We would like to thank Reviewer #1 for understanding the strength of our simple HSC culture system and interest to the immunology/hematopoiesis community. We also appreciate the suggested mechanistic studies of our whole BM cell cultures. To answer this question, we have performed new experiments to compare cell death and its effects after short-term culture of whole BM and c-Kit+ BM cells: Apoptosis assays and q-PCR gene expression analysis of senescence-associated genes, *p16^{INK4a}*, *p19^{Arf}* and *Trp53* (Figure 2E-F).

The frequency of Propidium Iodide+ Annexin V+ cells was higher in the whole BM cultures than in the c-Kit+ BM cultures on both Day 7 and Day 14 (Figure 2E). Expression of *p16^{INK4a}* and *p19^{Arf}* were also higher in the cultured whole BM cells than in the cultured c-Kit+ BM cells, likewise in both the bulk and the c-Kit+Sca1+Lineage- (KSL) cell fraction. Expression of *Trp53* was also predominantly higher in the whole BM than the c-Kit+ BM cultures in the bulk, but not in the KSL fraction (Figure 2F). Together these results suggest that the whole BM cells undergo more cell death than the c-Kit+ culture cells, and this correlated with reduced HSC activity within the culture.

Impressively, the authors show 30% long-term engraftment after transplanting 1 million culture-enriched CD117+ cell progeny into non-conditioned mice suggesting substantial ex vivo expansion, a result which was reproduced in a second experiment with genetically-engineered and in vitro selected cells. While novelty is somewhat limited due to the previous publication, a simplified protocol including a genetic engineering/selection step that yields up to 30% of long-term engraftment in non-conditioned mice is of significant interest to the community studying immunology & hematopoiesis. The following major points should be addressed in order to ensure robustness and reproducibility of the transplantation protocol:

1) The authors should indicate the “day 0 equivalent” number of KSL or Lin- cells that were transplanted in the various experiments, in order to allow for a better comparison between test-, control- and supporter cell dose. Absolute numbers of KSL or CD34-KSL cells in the mouse BM (Fig.1F, 2F) should be provided, if possible.

We now include this information the main text (page 3 line 3, page 3 line 19, page 4 line 17, and page 4 line 22). Unfortunately, we are not able to accurately estimate the absolute numbers of KSL or CD34⁺KSL in the recipient mice in Figures 1 and 2 -we can only provide frequencies as not all BM was collected/analyzed.

2) LT-HSC expansion from Kit⁺ cultures should be quantified according to state-of-the-art functional assays, namely limiting-dilution transplantation and/or clonality analysis of lentivirally-marked cells. Functional data are important, as immunophenotypes revealed in culture may not have the same significance as on fresh cells.

To provide a more precise quantification of functional expansion *ex vivo*, we performed limiting dilution assays of fresh and expanded c-Kit⁺ BM cells. This data is displayed in Figure 1I. We would like to note that these limiting dilution assays were performed against 1×10^6 whole BM competitor cells, a more stringent functional assay than the usual limiting dilution assays performed using 2×10^5 whole BM competitor cells. Within this assay, we estimate HSC frequency in the expanded cells at 1:2164 cells, whereas fresh c-Kit⁺ cells had an HSC frequency of 1:3971.

3) What is the “dose-response” relationship for non-conditioned transplants? Are there threshold levels for the number of LT-HSC units/total cells in order to achieve stable engraftment in the double digit % range? What is the mechanism of HSC exchange? Is it the stoichiometric ratio between endogenous HSC and transplanted HSC? Is the timing of physiologic HSC mobilization & homing (and therefore, the persistence of transplanted HSC in alternative niches) relevant, or can expanded HSC displace endogenous HSC from the bone marrow niche (producing a gain-of-function)? Is *ex vivo* culture essential, or can a “mega-dose” of fresh KSL cells provide equivalent levels of engraftment in the non-conditioned setting?

To answer this question, we performed limiting dilution transplantation assays into non-conditioned recipients using fresh and cultured cells (1×10^6 , 1×10^5 , and 1×10^4 cell doses tested). As showed in Figure 3D, equivalent doses of expanded cells show substantially higher engraftment. This is consistent with the expansion of functional HSPCs during culture. We believe this data is difficult to compare with the conventional limiting dilution assay, we therefore did not perform any estimate of stem cell frequency. Please note that we also performed limiting dilution transplantation assays into irradiated recipients (see Figure 1I for details), which also confirmed increased functional potential post-expansion.

4) How do the authors explain the discrepancy between the %GFP⁺ cells *in vitro* (close to 100%) and *in vivo* (around 30%) in Fig.4? Promoter silencing? Incomplete *in vitro* selection at the HSC (as opposed to the KSL) level?

To investigate this, we re-performed the experiment in both *in vitro* and *in vivo* settings using an optimized DOX protocol. In this repeat, EGFP-transduced c-Kit⁺ cells showed high EGFP expression in the DOX on state. We have added this data to Figure 4. We believe our new data resolves this concern.

Reviewer #2 (Remarks to the Author):

Ochi et al describe a novel strategy that allows enrichment of functional HSPCs without sorting, and the subsequent generation of bone marrow chimeras without host conditioning. Overall the approach may prove to be quite useful and overcome some technical hurdles associated with generation of BM chimeras. Moreover, achieving robust engraftment in non-irradiated recipients may enable new insights about hematopoiesis and the immune system in models that previously have been assessed using lethally irradiated recipients. However, the culture system has already been

published as well as the generation of bone marrow chimeras without irradiation. As such the conceptual advance is incremental. The study lacks several important controls to enable assessment of the actual efficiency of the system. The authors should further demonstrate how the system may benefit the study of hematopoiesis and immune system using models with biological effects, in addition to their GFP reporter. Finally, I have some serious concerns about the rigidity of the data analysis (see below).

We would like to thank Review #2 for their careful and constructive review of our manuscript. We hope that we have fully resolved all concerns in our revised manuscript (please see below).

Specific remarks:

Figure 1: This is mostly negative data and could be moved to the supplement

We have reworked our manuscript/figures so that **Figures 1 and 2** are now swapped in order to highlight our key data from the c-Kit⁺ BM cultures and moved some whole BM cell culture data to the supplement. We thank Reviewer #2 for this suggestion.

Figure 2: The degree of expansion appears quite modest compared to what was previously reported for purified HSCs. A precise quantification of the actual expansion would be necessary to understand this. The competitive transplants are difficult to understand as 1 million fresh c-kit⁺ cells appear to give compete equally to 1 million whole BM cells (around 50% chimerism). Would why are the c-kit⁺ cells dominating over the whole BM cells in the long term?

As we did not observe c-Kit⁺ cells dominating over the whole BM cells in this figure, we believe the question should read “**Would why are the c-kit⁺ cells not dominating over the whole BM cells in the long term?**”. We often observe that whole BM cells perform well in transplantation assays although, as the Reviewer alludes to, has lower HSC frequencies. To provide a more precise quantification of expansion, we performed limiting dilution assays of fresh and expanded c-Kit⁺ BM cells. This data is displayed in **Figure 1I**. We would like to note that these limiting dilution assays were performed against 1×10^6 whole BM competitor cells, a more stringent functional assay than the usual limiting dilution assays performed using 2×10^5 whole BM competitor cells. Within this assay, we estimate that the engraftable HSC frequency in the expanded cells as 1:2164 cells, whereas fresh c-Kit⁺ cells had an HSC frequency of 1:3971.

Figure 3: The level of engraftment from only 1 million expanded cells is quite remarkable. However, it would be essential to include a control with equivalent and higher numbers of non-cultured cells to understand the benefit of the culture system for non-irradiated recipients.

To answer this question, we performed limiting dilution transplantation analysis into non-conditioned recipients using fresh and cultured cells (1×10^6 , 1×10^5 , and 1×10^4 cell doses tested). As showed in **Figure 3D**, equivalent doses of expanded cells show substantially higher engraftment. This is consistent with the expansion of functional HSPCs during culture.

Figure 4: The authors should explain why the GFP induction only works in around 30% of the donor cells despite transduced cells being selected for by puromycin.

To investigate this, we re-performed the experiment in both in vitro and in vivo settings using an optimized DOX protocol. In this repeat, EGFP-transduced c-Kit⁺ cells showed high EGFP expression in the DOX on state. We have added this data to **Figure 4**. We believe our new data resolves this concern.

The authors should carefully reanalyze and reexamine their data and figures. Careful examination of the FACS plots in Figure 4A shows that the second and fourth plot are must be derived from the same sample but with more events shown in the fourth plot.

This may be a simple mistake but is very concerning and puts into question the rigor of the entire study and the analysis of the remainder of the data presented in the paper.

We thank Reviewer #2 for identifying this error in Figure 4 and apologize for this mistake when pasting our representative figures from FlowJo. To avoid any doubt, we repeated this entire experiment and now only include the new dataset within the manuscript (**Figure 4B**). We submit original individual flow cytometry data attached to the last page. We have also carefully checked all other datasets presented and are confident that all figures are correct.

Related to Figure 4D.

All samples were analyzed in triplicate.

REVIEWERS' COMMENTS

Reviewer #1 (Remarks to the Author):

The authors have performed new experiments that sufficiently address my initial concerns.

Bernhard Gentner

Reviewer #2 (Remarks to the Author):

The authors have performed extensive additional experiments and my concerns have been appropriately addressed.

Two minor details:

Figure 1L Could the authors extrapolate from these data what the total expansion of HSCs would be? With about four-fold expansion of c-kit+ cells and a two-fold increase of HSC frequency, it seems reasonable to estimate the total HSC expansion to be around 8-fold. Correct?

Figure 4H It would be more informative to see the percentage of GFP+ cells in this graph rather than MFI.

Reviewer #1 (Remarks to the Author):

The authors have performed new experiments that sufficiently address my initial concerns.
Bernhard Gentner

We would like to thank Reviewer #1 for kindly re-reviewing our manuscript.

Reviewer #2 (Remarks to the Author):

The authors have performed extensive additional experiments and my concerns have been appropriately addressed.

We would like to thank Reviewer #2 for re-reviewing our manuscript and their constructive comments.

Two minor details:

Figure 1L Could the authors extrapolate from these data what the total expansion of HSCs would be? With about four-fold expansion of c-kit⁺ cells and a two-fold increase of HSC frequency, it seems reasonable to estimate the total HSC expansion to be around 8-fold. Correct?

We have added this estimate to the main text (page 4, line 11).

Figure 4H It would be more informative to see the percentage of GFP⁺ cells in this graph rather than MFI.

We have updated Figure 4H as suggested.